# Hypertension prevalence in patients attending tertiary pain management services, a registry-based Australian cohort study

**Melita J. Giummarra**[1,2]*, **Hilarie Tardif**[3], **Megan Blanchard**[3], **Andrew Tonkin**[1], **Carolyn A. Arnold**[2,4]

**1** Department of Epidemiology & Preventive Medicine, School of Public Health and Preventive Medicine, Monash University, Melbourne, Victoria, Australia, **2** Caulfield Pain Management and Research Centre, Caulfield Hospital, Caulfield, Victoria, Australia, **3** Australian Health Services Research Institute, University of Wollongong, Wollongong, New South Wales, Australia, **4** Academic Board of Anaesthesia & Perioperative Medicine, School of Medicine Nursing & Health Sciences, Monash University, Clayton, Victoria, Australia

* melita.giummarra@monash.edu

**Data Availability Statement:** The authors do not have permission to share the data as they were provided specifically within the scope of the study protocol as approved by the ethics committee. The

## Abstract

Persistent pain and hypertension often co-occur, and share several biological and lifestyle risk factors. The present study aimed to provide insight into the prevalence of, and factors associated with, hypertension in the largest cohort of patients seeking treatment in 43 tertiary pain clinics in Australia. Adults aged > = 18 years registered to the electronic Persistent Pain Outcomes Collaboration registry between 2013 and 2018 were included if they had persistent non-cancer pain (N = 43,789). Risk Ratios (RRs) compared prevalence of self-reported hypertension with the general and primary care Australian populations, and logistic regression examined factors associated with hypertension. One in four (23.9%) patients had hypertension, which was higher than the Australian adult population (2014–15: RR = 5.86, 95%CI: 5.66, 6.06; 2017–18: RR = 9.40, 95%CI: 9.01, 9.80), and in primary care patients (2011–13: RR = 1.17, 95%CI: 1.15, 1.20). Adjusting for covariates, patients with higher odds of hypertension were older, lived in regions with higher socioeconomic disadvantage, had higher levels of BMI, were born outside the Oceania/Australasia region, and had comorbid arthritis, diabetes, or severe-extremely severe anxiety symptoms. Female patients and those with depression symptoms had lower adjusted odds. Unadjusted analyses showed an association between widespread pain, pain duration, pain severity and interference, and lower pain self-efficacy with hypertension; however, only pain severity remained significant in adjusted analyses. Hypertension was more prevalent in people with persistent pain than in the general community, was associated with more severe pain, and commonly co-occurred with pain-related impairments. Routine hypertension screening and treatment targeting shared mechanisms of hypertension and pain may improve treatment outcomes in the pain clinic setting.

authors are bound by a publishing agreement with the electronic Persistent Pain Outcomes Collaboration that legally prevents them from disseminating the raw study data. However, it is possible for external parties to request a copy of the same data used in this study through a request directly to the Data Access Working Group of the Electronic Persistent Pain Outcomes Collaboration at the University of Wollongong. Requests for these data would require independent ethics approval. Data access inquiries can be made to ePPOC via eppoc-uow@uow.edu.au.

**Funding:** This work was supported by an ARC DECRA fellowship (DE170100726) to MJG. The funder had no role in study design, data collection and analysis, decision to publish, or preparation of the manuscript.

**Competing interests:** The authors have declared that no competing interests exist.

## Introduction

Several studies have shown higher prevalence of hypertension (i.e., systolic blood pressure >140mmHg) in people with persistent pain [1, 2]. People with persistent pain typically have higher blood pressure both at rest and when experiencing a painful stimulus (e.g., in the cold pressor test) [1] and prevalence is higher in people with more severe persistent pain [3].

A number of factors have been found to increase the risk of developing both persistent pain and other chronic diseases [4], especially cardiovascular diseases [1, 5–8]. These include *psychological* factors, such as depression, anxiety and stress), *lifestyle* factors (e.g., obesity, low physical inactivity, deconditioning) and *social* factors (e.g., isolation, unemployment, lower education) [9, 10]. Moreover, *biological* factors probably play a role given that people with persistent pain have lower baroreceptor sensitivity, and diminished inhibitory engagement of the parasympathetic nervous system at rest [11–13]. Some factors such as smoking, low levels of physical activity and obesity probably play a causal role in blood pressure and cardiovascular disease development, possibly independent of pain severity [14], due to their impact on vital homeostatic processes and organ function. Other factors like anxiety, however, probably have an indirect association via disruption in stress regulation systems [15] and decreased parasympathetic tone [13].

Understanding the prevalence of hypertension, and demographic, clinical and pain-related features associated with hypertension in patients with persistent pain will provide an important step in the future development and implementation of tailored interventions for patients seeking treatment for pain in outpatient pain management settings. While the co-occurrence of persistent pain and hypertension has been examined in several community [1, 2] and primary care cohorts [16], few studies have focused on patients with pain that is severe or disabling enough to warrant treatment in a multidisciplinary pain management service. In fact, only one study published nearly 15 years ago has examined the prevalence and clinical manifestation of hypertension in a small sample of 300 patients attending a tertiary pain management service in the the United States of America, which found that 39% of patients had hypertension [8]. The present study therefore aimed to provide an updated robust insight into the prevalence of hypertension in a large cohort of patients referred to outpatient pain management clinics in Australia, and to investigate the demographic, health and pain-related characteristics associated with having hypertension.

## Methods

### Setting

In Australia, pain management services are specialist sub-acute ambulatory care services located primarily in public and private hospital settings. Most clinics comprise medical staff (pain specialists, psychiatry, anaesthetists, rehabilitation medicine consultants, and general practitioners) and senior allied health and nursing staff (physiotherapists, occupational therapists, psychologists, and clinical and research nurses). In 2013, the Australian Health Services Research Institute, University of Wollongong, established the electronic Persistent Pain Outcomes Collaboration (ePPOC) to facilitate outcome assessment and service benchmarking. The design, procedures and characteristics of ePPOC are fully described elsewhere [17]. Under ePPOC, patients referred to specialist outpatient pain management clinics complete a standard battery of outcome measures at referral or prior to commencing treatment, and at set follow-up intervals according to a defined protocol.

### Participants

All new patients entered in ePPOC up to September 2018, aged 18 years and over, who were seeking treatment for non-malignant persistent pain and responded to the primary outcome

measure (patient reported hypertension status) were included. Data obtained in the initial patient questionnaire completed at referral or commencement of treatment were used for this study. The dataset was screened for multiple referrals for the same patient, and only data from the most recent referral were extracted for analysis.

## Materials and procedures

Questionnaires were completed by patients and entered into the purpose-built software (epi-Centre) by staff at the respective pain management clinic. The study had low risk approval from the Alfred Health Human Research Ethics Committee for the analysis of fully deidentified data. Participants contribute their data as part of their clinical episode, and while they do not provide written consent for the use of the data in individual studies they are informed that their data will be used for service benchmarking, and that deidentified data may be used for research.

**Demographics.** Patient age, sex, and country of birth were used to characterise individual-level demographics. Country of birth was classified into regions according to the Standard Australian Classification of Countries [18], and summarised as Oceania and Antarctica, North-West Europe, Southern and Eastern Europe, North Africa and the Middle East, South-East Asia, North-East Asia, Southern and Central Asia, Americas, and Sub-Saharan Africa. Area level socioeconomic position was determined from patient residential postcode using the Index of Relative Socio-Economic Advantage and Disadvantage (IRSAD) decile [19], which were summarised into quintiles with lower scores indicating higher disadvantage. The IRSAD rankings are determined from typical education, employment and family structure in all Australian postal codes from the National Census of Population and Housing in 2011, and each area is ranked nationally.

**Health and comorbidities.** Body Mass Index (BMI) was calculated from patient height (cm) and weight (kg), and classified into World Health Organization [20] ranges (kg/m$^2$) of underweight (<18.5), normal weight (18.5 to <25), overweight (25 to <30), Obese Class I (30 to <35), Obese Class II (35 to <40) and Obese Class III (>40). Medical conditions and comorbidities were recorded in response to the question "do you have any of the following medical conditions?" which was followed by a list of conditions that included high blood pressure, arthritis (rheumatoid or osteoarthritis), and diabetes. The self-reported data are broadly consistent with population-based registry procedures for assessing the presence of comorbid conditions. Objective measures of blood pressure, or prescription antihypertensives were not available.

**Pain.** The Brief Pain Inventory (BPI) was used to measure pain severity and interference in the previous week [21]. Participants completed 11-point numerical rating scales for the intensity of pain (right now, on average, at worst and least; 0 "no pain", 10 "Pain as bad as you can imagine"), and its impact on daily functions, including general activity, mood, walking ability, normal work, relationships with other people, sleep, and enjoyment of life (0 "does not interfere", 10 "completely interferes"). Severity and interference scores were calculated according to the scoring guidelines [22], which showed high levels of internal consistency in this sample (Severity: Cronbach α = 0.88; Interference: Cronbach α = 0.89). Pain severity and interference were classified as low (0 to 3), moderate (4 to 6) and high (7 to 10) [23, 24].

Participants reported which body regions were affected by pain on a body map [25]. Widespread pain was classified according to the Widespread Pain Index [26], which involves summing the number of sites of pain across the neck, upper back, lower back, chest, abdomen, and the left and right side of the: jaw, shoulder girdle, upper arm, lower arm, hip/buttock/trochanter, upper leg and lower leg. As the ePPOC body map coded for pain in the head, but not in

the left/right cheek, the total possible score for the WPI was 18 instead of 19. Participants were classified as having widespread pain if they had a score $> = 7$, based on recommendations of the 2010 American College of Rheumatology criteria [27].

The Pain Self-Efficacy Questionnaire (PSEQ)[28] is a self-report measure of confidence in performing everyday tasks, despite being in pain. It comprises 10 items such as "I can enjoy things, despite the pain" that are rated on a scale from 0 to 6, with higher scores indicating greater confidence. Ratings are summed to produce a total score ranging from 0 to 60 with higher scores indicated better self-efficacy. Scores were classified as indicating severe impairment ($<$20), moderate impairment (20–30), mild impairment (31–40) and low impairment ($>$40) [28]. Cronbach's α for this sample was 0.93.

The Pain Catastrophizing Scale (PCS) [29] is a self-report measure of catastrophic thoughts and feelings (e.g., rumination, magnification or helplessness) about pain, which comprises 13 items that are rated on a 5-point Likert scale. The PCS total score ranges from 0 to 52, and scores can be classified as clinically normal (0–19), high (20–29) and clinically severe ($> = 30$) [30]. Cronbach's α for the total scale in this sample was 0.95.

**Mood.**   The 21-item version of the Depression Anxiety Stress Scale (DASS-21) was used to characterise depression and anxiety symptom severity [31]. The DASS has been shown to validly measure the dimensions of depression (Cronbach $\alpha = 0.93$, in this sample), and anxiety (Cronbach $\alpha = 0.86$, in this sample) in large non-clinical samples [32], and in patients with persistent pain [33, 34]. Each subscale score was doubled to enable use of the DASS-42 severity classifications for depression: normal (scores: 0–9), mild (scores 10–13), moderate (scores 14–20), severe (21–27) and very severe ($> = 28$); and anxiety: normal (scores: 0–7), mild (scores 8–9), moderate (scores 10–14), severe (15–19) and very severe ($> = 20$).

**Hypertension prevalence data sources.**   The prevalence rate of self-reported hypertension in the ePPOC cohort was compared with the Australian population (aged $> = 18$ years), and an Australian primary care population (aged $> = 24$ years). The prevalence in the Australian population was taken from the 2014 and 2017 Australian Bureau of Statistics (ABS) National Health Survey (NHS) data [35, 36]. The NHS is a purposely sampled survey intended to be representative of persons living in the general community in Australia; however, people living in institutions or in remote regions are not sampled. The NHSs recorded the prevalence of long-term health conditions (i.e., a condition that had lasted, or will last, for at least 6-months) in 2011–12, and 2017–18, respectively. We compared self-reported hypertension in the present ePPOC cohort with the prevalence of self-reported hypertension from the NHS interview. The NHS interview recorded hypertension status in response to the questions "Have you ever been told by a doctor or nurse that you have . . . condition?" and "Is this condition current and/or long term ($>$6 months)?". The prevalence of hypertension in Australian primary care patients was taken from a sentinel study by Ghosh, Charlton [37], which examined the prevalence of chronic conditions diagnosed and recorded by a general practitioner from all patient interactions over a two year period (September 2011 to September 2013) from 17 general practices in a single health district in New South Wales, Australia (N = 103,917). The crude, and age-adjusted prevalence in the ePPOC cohort were compared with the crude and age-adjusted prevalence of hypertension in the study by Ghosh, Charlton [37].

## Statistical analysis

The data were analysed using Stata Version 14.0. Significance was determined when the confidence interval (CI) for risk ratios (RRs), or odds ratio (ORs), did not include 1.00. BPI severity and interference subscales were calculated if patients completed all four severity ratings, and a minimum of four interference items. Subscale scores were calculated for responses given to

items on the DASS anxiety subscale, DASS depression subscale, PSEQ or PCS; however, participants missing more than one item were coded as missing for that measure consistent with previous recommendations [38]. Responses to a single missing item were not imputed to generate summary scores. BMI was calculated after applying sex-stratified plausible weight and height values based on criteria published from a previous Australian epidemiological project, the Household, Income and Labour Dynamics in Australia (HILDA) study(height: men = 130 to 229cm; women = 110 to 210cm; weight: men = 35 to 300kg; women = 25 to 300kg) such that height and weight falling outside of those ranges were coded as missing due to a probable data entry error [39].

The RR and corresponding 95% CIs were calculated to determine the difference in risk of hypertension in the ePPOC cohort compared with the general population, and the primary care population.

The association between demographic, health and pain-related characteristics and hypertension were examined using logistic regression. The ORs and 95% CIs were calculated, including ORs without adjusting for covariates, and ORs covarying for all demographic, clinical and pain characteristics. The fully adjusted model estimated missing data from the covariates using multiple imputation by chained equations, which imputes one variable at a time, conditional on the other variables included in the multivariable model, through multiple iterations [40]. Twenty imputed datasets were produced and combined using Rubin's rules [41]. Factors were retained in multivariable analyses if they had an unadjusted p-value of <0.20 in accordance with previous recommendations [42]. Factors were considered based on prior clinical studies examining factors associated with hypertension, and included age, sex, birth region, socioeconomic disadvantage (IRSAD quintiles), BMI, arthritis, diabetes, anxiety symptom severity, depression symptom severity, pain source, pain duration, presence of widespread pain, pain severity, pain interference pain self-efficacy and pain catastrophizing. Multicollinearity was assessed, and no issues were identified, with an overall Variance Inflation Factor of <2.50. Sensitivity tests were performed to examine whether the results from the logistic regression were robust. For the first sensitivity test, the cohort was randomly split into two groups (using the rbinomial stata command). The two groups were first compared to ensure that they did not differ in relation to the prevalence of hypertension. Then the fully adjusted logistic regression was re-executed in each group, and the coefficients in each sub-sample were statistically compared using the suest Stata function. For the second sensitivity test, ORs were visually compared between the imputed dataset and the same logistic regression that only included participants with complete datasets.

Finally given that hypertension is typically associated with disability and lower activity levels, differences in each pain interference domain between those with and without hypertension were examined using simple inferential statistics (t-tests and hedges *g*). Differences in interference levels were considered to show minimal clinically significant differences if they exceeded 10%, moderately different if they exceeded 30%, and substantially different if they exceeded 50% [43].

## Results

### Cohort overview

Data were provided by 43 pain management services in four Australian states. During the study period 52,873 patients at Australian pain management services were entered into epiCentre. A total of 45,272 (85.6%) patients completed initial questionnaires during this period, of whom 43,789 (96.7%) were eligible for inclusion, Fig 1.

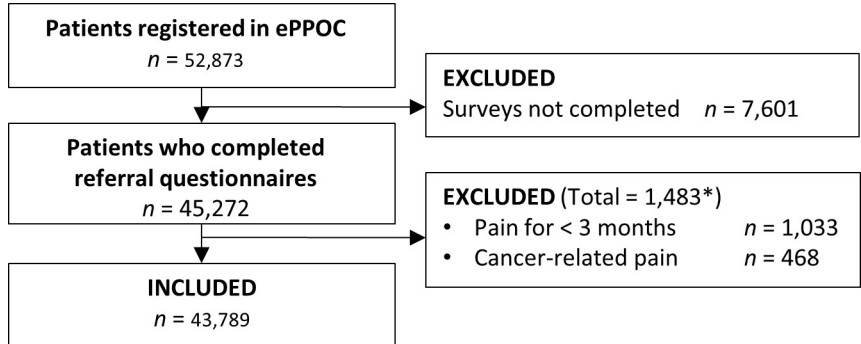

**Fig 1. STROBE recruitment chart.** *Notes*: 18 patients met multiple exclusion criteria (\*). Age: mean = 52.17 years, SD = 15.59.

The cohort were, on average, 52 years of age (*sd* = 15.59, 95% CI: 27 to 79 years), and were predominantly female (*n* = 25,342, 57.9%). Most patients were seeking treatment for pain that commenced after an injury (*n* = 20,185, 48.0%); however, a large proportion reported that their pain had no obvious cause (*n* = 7,583, 18.0%), commenced after surgery (*n* = 4,139, 9.8%) or illness (n = 4,795, 11.4%), or had some other cause (*n* = 5,392, 12.8%); missing *n* = 1,695. Almost two thirds of patients reported that they had pain localised in at least one region of the back (*n* = 25,176, 57.5%). Most patients with pain in the arms or legs reported pain that affected both sides, and less than ten percent of the cohort reported pain that pain affected one side of specific body regions only, Fig 2. More than two thirds (*n* = 30,478, 69.6%) of patients reported three or more sites of pain, and 14,401 (32.9%) reported seven or more sites of pain. Additional demographic characteristics are reported in Table 1.

## Prevalence of hypertension in the ePPOC cohort

A total of 10,472 (23.9% crude prevalence; 20.0% age-adjusted prevalence) patients registered to ePPOC reported having high blood pressure in their referral questionnaires. Hypertension was significantly more prevalent in the ePPOC cohort than the general population in 2014–15 (RR = 5.86, 95%CI: 5.66 to 6.06), 2017–18 (RR = 9.40, 95%CI: 9.01 to 9.80), and compared with patients attending primary care practices in a single health district in NSW, Australia (2011–13: RR = 1.17, 95%CI: 1.15 to 1.20). The age-adjusted rates of hypertension were higher than the unadjusted rates in the ePPOC cohort relative to primary care attendees (RR: 1.68, 95%CI: 1.64, 1.72).

## Patient characteristics associated with hypertension

Older age was consistently associated with higher odds of hypertension, whereas women had 12% lower adjusted odds of having hypertension. People living in neighbourhoods with higher levels of socioeconomic disadvantage had 10–12% higher adjusted odds of having hypertension, and people who were born outside of the Oceania region had 14% higher adjusted odds of hypertension.

Increasing BMI levels had markedly higher adjusted odds of having hypertension, from 60% higher odds for overweight patients up to 3.89-fold higher odds for those with Class III obesity. The adjusted odds of hypertension was 50% higher in people with arthritis, and 2.42-fold higher in people with diabetes. Severe and extremely severe anxiety symptoms increased the adjusted odds of hypertension by 26%; however, depression symptoms were

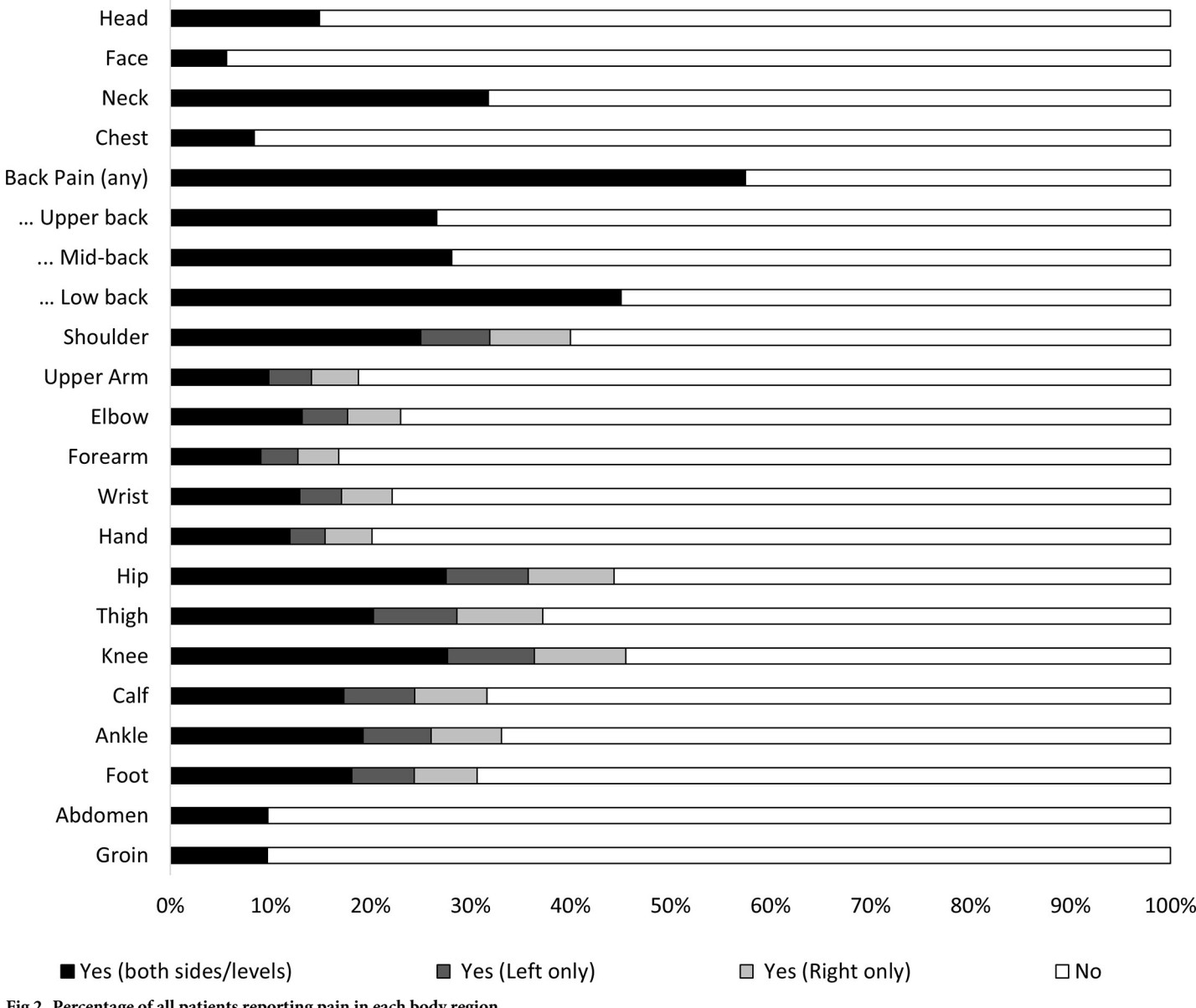

**Fig 2. Percentage of all patients reporting pain in each body region.**

associated with lower odds of having hypertension by 9–13%, but only when adjusting for all other demographic and clinical covariates.

Hypertension was more prevalent in people whose pain was began after surgery, with 10% higher adjusted odds compared with people whose pain began after injury. While widespread pain had a crude association with hypertension, with 23% higher odds in people with 3–6 sites of pain and 48% higher odds in those with seven or more sites of pain, this association was no longer significant when adjusting for other demographic, clinical and pain-related characteristics. Moderate and high levels of pain severity increased the odds of having hypertension by 23–47%; however the magnitude of these associations was reduced to 15–17% when adjusting for all demographic, clinical and pain characteristics.

## Sensitivity analyses

Sensitivity analyses showed that the overall results were consistent between two randomly generated sub-groups, $\chi^2(42) = 27.86$, p = 0.95. The findings were consistent for most variables when comparing the AOR for the analyses that included multiple imputation (N = 43,789) and those that had complete datasets (N = 21,653), S1 Table. The exceptions were that the complete dataset showed a significantly increased odds of hypertension only in the most disadvantaged neighbourhood versus the two most disadvantaged neighbourhoods in the imputed dataset, and in people who had no obvious cause for their pain, which did not significantly differ from the reference group (pain after injury) in the imputed dataset. On the contrary, there was only reduced odds of having hypertension for people with severe/extremely severe depression in the complete case analysis versus reduced odds for both moderate and severe/extremely severe depression groups in the imputed dataset. There was reduced odds of hypertension for people in the underweight BMI range, which was not significant in the imputed dataset. While the direction of the associations between hypertension and pain severity, age, and BMI were in the same direction in both analyses, the magnitude of the associations were slightly stronger in the complete case analysis than in the imputed dataset.

## Hypertension and interference of pain with activity, mood and enjoyment of life

Very small differences in pain interference were reported by patients with hypertension compared with those without hypertension with the exception of walking ability which had a mean difference of 0.69; however, this difference was not clinically significant, Fig 3, and S2 Table.

## Discussion

This large pain registry-based study found that one in four patients attending pain clinics had hypertension. Hypertension prevalence in these patients was 5.6 to 9.8 times higher than the Australian general population, and 1.2 times higher than the Australian primary care setting, and 1.7 times higher when adjusting for age in the clinical samples. While the direction of these findings is consistent with previous studies, the prevalence of 24% in this large and clinically representative cohort was much lower than the 39% prevalence found in a much smaller cohort of 300 patients attending an American pain clinic [8]. This suggests that previous studies may have overstated the prevalence of hypertension in people living with persistent, disabling pain who present to a pain clinic for treatment. However, it is possible that differences in the demographic and clinical characteristics between Australian and American patients may explain the differences in the hypertension prevalence. Hypertension was associated with advancing age, higher BMI, greater socioeconomic disadvantage, being an immigrant (i.e., being born outside of the Oceania region), having arthritis or diabetes, and moderate to extremely severe anxiety symptoms. Depression, however, was not associated with hypertension in unadjusted analyses, and appeared to be protective in the adjusted analyses.

The relative risk of having hypertension in the pain clinic sample was higher when compared with the sample representative of the Australian population and a primary care cohort, although the hypertension prevalence in the pain clinic sample was much more similar to the clinical sample relative to the general population. These observations are understandable given that people seeking treatment in both a primary care clinic and a pain clinic are likely to be older and to have more comorbid conditions, especially conditions requiring treatment with medications like hypertension, than people drawn from the general population. While we could not examine covariate adjusted prevalence of hypertension between the pain clinic

**Table 1. Association between demographic, clinical and pain characteristics with hypertension (N = 43,789) adjusting for all demographic and clinical characteristics.**

| | Total | No hypertension | Hypertension | | |
|---|---|---|---|---|---|
| | N (%) | N (%) | N (%) | OR (95% CI) | AOR (95% CI) |
| Sex [c] | | | | | |
| Male | 18,405 (42.1) | 14,098 (42.4) | 4,307 (41.2) | Reference | Reference |
| Female | 25,342 (57.9) | 19,191 (57.6) | 6,151 (58.8) | 1.05 (1.00, 1.10) | **0.88 (0.83, 0.93)** |
| Age [d] | | | | | |
| 18 to 24 years | 1,458 (3.3) | 1,428 (4.3) | 30 (0.3) | Reference | Reference |
| 25 to 34 years | 4,733 (10.8) | 4,523 (13.6) | 210 (2.0) | **2.21 (1.50, 3.25)** | **1.87 (1.27, 2.75)** |
| 35 to 44 years | 7,991 (18.2) | 7,219 (21.7) | 772 (7.4) | **5.09 (3.52, 7.36)** | **3.74 (2.58, 5.41)** |
| 45 to 54 years | 10,777 (24.6) | 8,706 (26.1) | 2,071 (19.8) | **11.32 (7.86, 16.31)** | **7.40 (5.13, 10.67)** |
| 55 to 64 years | 9,340 (21.3) | 6,298 (18.9) | 3,042 (29.0) | **22.99 (15.97, 33.09)** | **13.70 (9.50, 19.76)** |
| 65 to 74 years | 5,516 (12.6) | 3,116 (9.4) | 2,400 (22.9) | **36.66 (25.44, 52.84)** | **20.29 (14.04, 29.34)** |
| 75 to 84 years | 3,181 (7.3) | 1,605 (4.8) | 1,576 (15.0) | **46.74 (32.34, 67.55)** | **27.53 (18.96, 39.97)** |
| > = 85 years | 791 (1.8) | 420 (1.3) | 371 (3.5) | **42.05 (28.54, 61.95)** | **30.65 (20.63, 45.55)** |
| Birth Region [e] | | | | | |
| Oceania and Antarctica | 30,776 (73.7) | 24,025 (75.8) | 6,751 (67.2) | Reference | Reference |
| Other | 10,979 (26.3) | 7,691 (24.2) | 3,288 (32.8) | **1.52 (1.45, 1.60)** | **1.14 (1.08, 1.21)** |
| IRSAD Quintiles [f] | | | | | |
| 5 (lowest disadvantage) | 9,420 (22.8) | 7,279 (23.2) | 2,141 (21.5) | Reference | Reference |
| 4 | 7,404 (17.9) | 5,684 (18.1) | 1,720 (17.3) | 1.03 (0.96, 1.11) | 1.05 (0.97, 1.14) |
| 3 | 9,205 (22.3) | 6,961 (22.2) | 2,244 (22.5) | **1.10 (1.02, 1.17)** | 1.08 (1.00, 1.16) |
| 2 | 8,196 (19.8) | 6,162 (19.6) | 2,034 (20.4) | **1.12 (1.05, 1.20)** | **1.10 (1.01, 1.19)** |
| 1 (highest disadvantage) | 7,123 (17.2) | 5,299 (16.9) | 1,824 (18.3) | **1.17 (1.09, 1.26)** | **1.12 (1.03, 1.22)** |
| Body Mass Index [g] | | | | | |
| Normal weight | 8,751 (27.2) | 680 (2.8) | 1,098 (14.7) | Reference | Reference |
| Underweight | 756 (2.4) | 7,653 (31.0) | 76 (1.0) | 0.78 (0.61, 1.00) | 0.81 (0.63, 1.04) |
| Overweight | 9,975 (31.0) | 7,828 (31.7) | 2,147 (28.7) | **1.91 (1.77, 2.07)** | **1.60 (1.47, 1.74)** |
| Obese, Class I | 6,711 (20.9) | 4,741 (19.2) | 1,970 (26.4) | **2.90 (2.67, 3.14)** | **2.30 (2.09, 2.52)** |
| Obese, Class II | 3,343 (10.4) | 2,196 (8.9) | 1,147 (15.3) | **3.64 (3.31, 4.00)** | **2.90 (2.62, 3.21)** |
| Obese, Class III | 2,601 (8.1) | 1,566 (6.3) | 1,035 (13.8) | **4.61 (4.16, 5.10)** | **3.89 (3.47, 4.36)** |
| Comorbidities | | | | | |
| Arthritis | 14,496 (33.1) | 9,372 (28.1) | 5,372 (51.3) | **2.86 (2.72, 2.98)** | **1.50 (1.42, 1.59)** |
| Diabetes [h] | 4,881 (12.7) | 2,274 (7.9) | 2,607 (27.3) | **4.38 (4.12, 4.66)** | **2.42 (2.26, 2.60)** |
| Anxiety [i] | | | | | |
| Normal/mild | 17,030 (40.6) | 12,898 (40.8) | 3,731 (37.5) | Reference | Reference |
| Moderate | 7,637 (18.2) | 5,780 (18.3) | 1,857 (18.7) | **1.11 (1.04, 1.18)** | 1.07 (1.00, 1.16) |
| Severe/Extremely severe | 17,266 (41.2) | 12,912 (40.9) | 4,354 (43.8) | **1.17 (1.11, 1.23)** | **1.26 (1.17, 1.36)** |
| Depression [j] | | | | | |
| Normal/mild | 14,598 (34.7) | 10,741 (33.9) | 3,462 (34.7) | Reference | Reference |
| Moderate | 7,718 (18.3) | 5,899 (18.6) | 1,819 (18.2) | 0.96 (0.90, 1.02) | **0.91 (0.84, 0.98)** |
| Severe/Extremely severe | 19,777 (47.0) | 15,073 (47.5) | 4,704 (41.7) | 0.97 (0.92, 1.02) | **0.87 (0.80, 0.95)** |
| Pain Source [k] | | | | | |
| Post-injury | 20,185 (48.0) | 16,017 (50.1) | 4,168 (41.2) | Reference | Reference |
| Post-surgery | 4,139 (9.8) | 2,975 (9.3) | 1,164 (11.5) | **1.50 (1.39, 1.62)** | **1.10 (1.01, 1.20)** |
| Related to illness | 4,795 (11.4) | 3,500 (10.9) | 1,295 (12.8) | **1.42 (1.32, 1.53)** | 1.09 (1.00, 1.18) |
| No obvious cause | 7,583 (18.0) | 5,596 (17.5) | 1,987 (19.6) | **1.36 (1.28, 1.45)** | 1.06 (0.99, 1.14) |
| Other cause | 5,392 (12.8) | 3,891 (12.2) | 1,501 (14.8) | **1.48 (1.38, 1.59)** | 1.08 (1.00, 1.17) |

*(Continued)*

**Table 1.** (Continued)

| | Total | No hypertension | Hypertension | | |
|---|---|---|---|---|---|
| | N (%) | N (%) | N (%) | OR (95% CI) | AOR (95% CI) |
| Pain Duration [l] | | | | | |
| 3 to 12 months | 5,681 (13.6) | 4,499 (14.3) | 1,182 (11.7) | Reference | Reference |
| 12 to 24 months | 6,410 (15.4) | 5,074 (16.1) | 1,336 (13.2) | 1.00 (0.92, 1.09) | 1.02 (0.93, 1.13) |
| 2 to 5 years | 10,095 (24.2) | 7,860 (24.9) | 2,235 (22.1) | **1.08 (1.00, 1.17)** | 1.06 (0.97, 1.16) |
| > 5 years | 19,508 (46.8) | 14,136 (44.8) | 5,372 (53.1) | **1.45 (1.35, 1.55)** | 1.09 (1.00, 1.18) |
| Widespread Pain Index | | | | | |
| < 3 sites | 13,311 (30.4) | 10,225 (30.7) | 3,086 (29.5) | Reference | Reference |
| 3–6 sites | 16,077 (36.7) | 12,275 (36.8) | 3,802 (36.3) | **1.23 (1.12, 1.35)** | 1.00 (0.94, 1.06) |
| 7+ sites | 14,401 (32.9) | 10,817 (32.5) | 3,584 (34.2) | **1.48 (1.35, 1.63)** | 1.06 (0.99, 1.14) |
| Pain Severity [m] | | | | | |
| Low | 3,105 (7.5) | 2,544 (7.9) | 597 (6.1) | Reference | Reference |
| Moderate | 21,259 (51.4) | 16,730 (52.0) | 4,827 (49.1) | **1.23 (1.22, 1.36)** | **1.15 (1.03, 1.29)** |
| High | 16,962 (41.1) | 12,873 (40.0) | 4,410 (44.8) | **1.47 (1.34, 1.62)** | **1.17 (1.04, 1.31)** |
| Pain Interference [n] | | | | | |
| Low | 3,059 (7.2) | 2,356 (7.3) | 703 (6.9) | Reference | Reference |
| Moderate | 14,159 (33.4) | 10,761 (33.4) | 3,398 (33.5) | 1.03 (0.97, 1.08) | 1.00 (0.89, 1.12) |
| High | 25,128 (59.3) | 19,074 (59.3) | 6,054 (59.6) | **1.10 (1.04, 1.16)** | 0.99 (0.88, 1.12) |
| Pain Self-Efficacy [o] | | | | | |
| Low impairment | 3,739 (8.9) | 2,798 (8.7) | 941 (9.3) | Reference | Reference |
| Mild impairment | 5,393 (12.8) | 4,012 (12.5) | 1,381 (13.7) | 1.02 (0.93, 1.13) | 1.07 (0.96, 1.19) |
| Moderate impairment | 10,945 (26.0) | 8,378 (26.2) | 2,567 (25.5) | **0.91 (0.84, 0.99)** | 1.00 (0.90, 1.10) |
| Severe impairment | 22,003 (52.3) | 16,815 (52.5) | 5,188 (51.5) | 0.92 (0.85, 0.99) | 1.02 (0.92, 1.13) |
| Pain Catastrophizing [p] | | | | | |
| Clinically normal | 10,589 (25.8) | 8,036 (25.7) | 2,553 (25.9) | Reference | Reference |
| High | 8,604 (20.9) | 6,616 (21.2) | 1,988 (20.2) | 0.95 (0.88, 1.01) | 1.01 (0.93, 1.09) |
| Clinically elevated | 21,918 (53.3) | 16,611 (53.1) | 5,307 (53.9) | 1.01 (0.95, 1.06) | 0.99 (0.91, 1.07) |

*Abbreviations*: AOR = Adjusted Odds Ratio, IRSAD = Index of Relative Social Advantage and Disadvantage.

Missing data

[c] *n* = 42

[d] *n* = 2

[e] *n* = 2,034

[f] *n* = 2,441

[g] *n* = 11,652

[h] *n* = 5,341

[i] *n* = 2,257

[j] *n* = 2,091

[k] *n* = 1,695

[l] *n* = 2095

[m] *n* = 2,463

[n] *n* = 1,443

[o] *n* = 1,709

[p] *n* = 2,678.

*Notes*: AOR adjusted for all demographic and clinical characteristics associated with hypertension (p<0.20), with multiple imputation with chained equations to estimate missing covariate data.

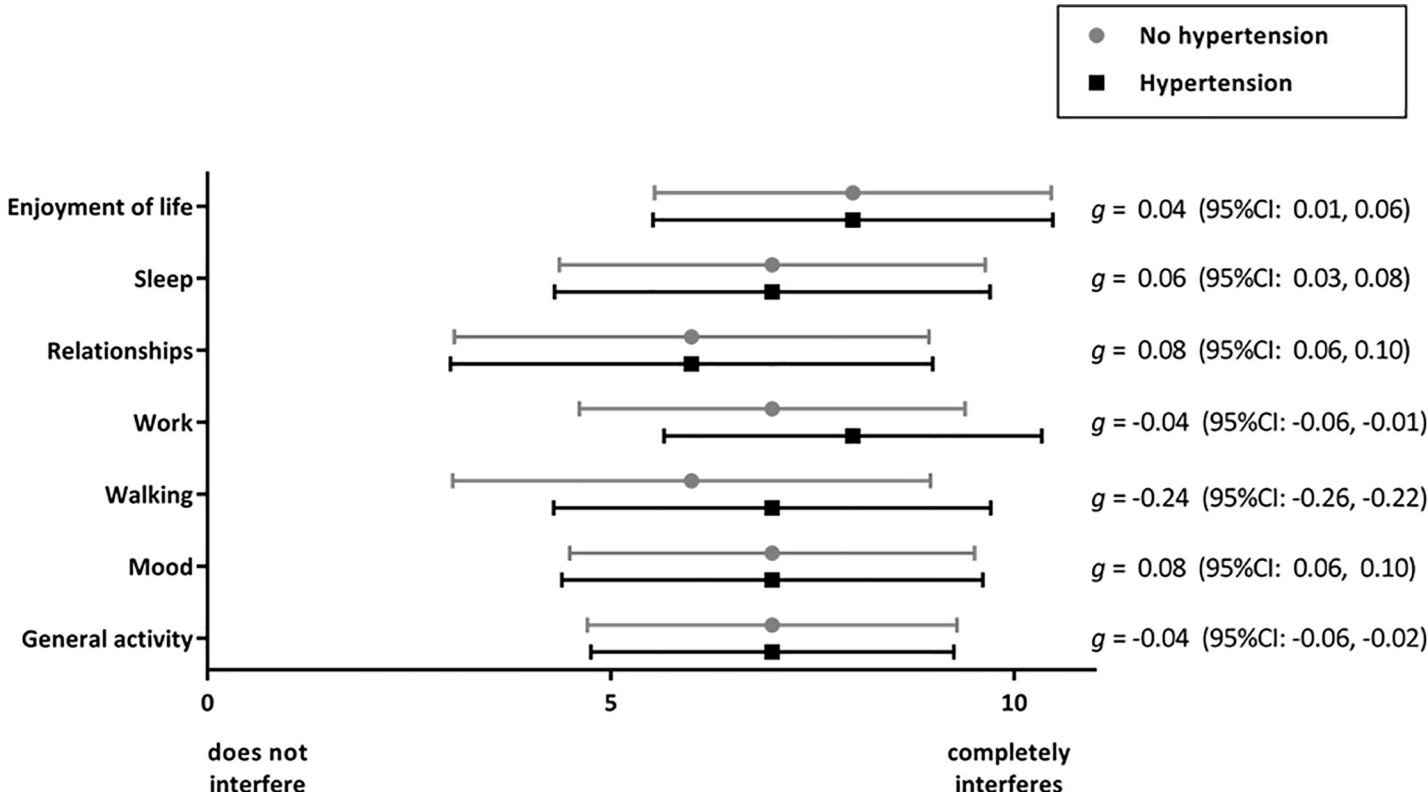

**Fig 3. Mean (standard deviation) levels of pain interference between patients with hypertension and without hypertension, raw values are available in S2 Table.**

cohort and the NHS and primary care samples, we could contrast the age-adjusted prevalence of hypertension between the clinical samples. These analyses showed that people attending a pain clinic had an even higher prevalence of hypertension compared with patients in a primary care setting when adjusting for patient age. This highlights the importance of accounting for age when examining the prevalence of conditions like hypertension that are known to increase in prevalence across the lifespan.

Hypertension was associated with several pain-related characteristics, particularly having pain that was related to illness, having pain for more than five years, pain of moderate or high severity, widespread pain, higher pain interference, lower pain self-efficacy and higher pain catastrophizing. However, when adjusting for all covariates hypertension was only associated with moderate to high pain severity. The other pain-related factors probably did not remain uniquely associated with hypertension when considered together given that they covary with each other and with other pertinent demographic characteristics. Patients with hypertension reported higher levels of interference of pain with their walking ability, general activity, and work; however, the magnitude of these differences was not clinically significant, and other factors probably have a stronger association with pain interference than blood pressure. A recent study found that the association between pain and blood pressure appeared to act via shared risk factors for both pain and cardiovascular disease, especially obesity [14], which may explain why most pain-related characteristics were no longer associated with hypertension in the fully adjusted analyses.

The present finding that hypertension is positively associated with the severity of persistent pain is consistent with previous studies in people attending primary care services in the United Kingdom [3], and patients attending an American pain clinic [8]. Not surprisingly, factors that

had the strongest association with hypertension (i.e., the greatest increase in odds) were attributable to life stage (i.e., age), and health and lifestyle factors (i.e., BMI, diabetes), which are known to have causal biological impacts on blood pressure [44]. Given that additional conditions like arthritis and diabetes, and poorer general health with high BMI, were associated with higher odds of having hypertension, multi-morbidity may be especially problematic in patients with both persistent pain and hypertension.

The protective effect of depression was not expected given that previous studies have shown that depression is positively associated with cardiovascular disease [45]. However, several studies have shown that there is limited evidence of a causal association between depression and hypertension [46], and even some evidence of a negative association between depression and blood pressure in community samples [47, 48]. One prospective study found that *low* blood pressure increased the incidence of later depression suggesting that blood pressure may play a causal role in depression [49], perhaps via enhanced fatigue and somatic symptoms. Further research is required to replicate the present findings in people living with persistent pain.

Biological models suggest that the higher prevalence of hypertension in people with hypertension may be due to insufficient inhibition and heightened facilitation in shared ascending/descending networks that govern both nociception and blood pressure [50]. In particular, people with persistent pain have been found to have lower baroreceptor sensitivity [12], suggesting that baroreflexes have reduced control over cardiovascular functions in people with persistent pain. Recent meta-analyses have shown that people with persistent pain, especially those with conditions involving widespread pain [51], have diminished capacity to engage inhibitory parasympathetic cardiovascular activity [13]. More recently, diminished activity in the parasympathetic nervous system has been found to mediate the association between persistent pain and hypertension [11].

As patients who have comorbid persistent pain and hypertension are at much higher risk of other poor health outcomes, including premature mortality [5, 52], routine screening and tailored treatments may be required to address the common biological and lifestyle risk factors for both conditions. The most effective management of these chronic conditions probably requires interdisciplinary care to ensure that patients do not only receive specialist pain management, but concurrent and integrated treatment of their comorbid conditions. This poses a challenge in many health systems that often have "siloed" approaches to chronic disease management.

## Future research

Further research is required to develop and evaluate the impact of interventions that target pain, hypertension, and the common lifestyle and health risk factors for both conditions, on clinical outcomes. Collaborative care has been identified as an emerging priority that may lead to better management of shared health and lifestyle factors associated with a range of chronic conditions. Moreover, specific lifestyle interventions and pharmacological treatments for hypertension [53], or transdisciplinary interventions targeting lifestyle to improve diet, activity, general health and psychological wellbeing [54] may improve both pain and cardiovascular outcomes. Cognitive-behavioural and lifestyle-focussed interventions can successfully improve activity capacity and fitness, and reduce activity avoidance [55], while also reducing excess body weight [56], through gradual increases in physical therapy and activity pacing. These treatments are generally safe for patients with persistent pain cardiovascular disease risks like hypertension, and are likely to have wide-ranging health benefits including improved exercise tolerance, reduced systolic blood pressure, improved quality of life [57], and reduced pain-related disability [58, 59]. Several complementary rehabilitation therapies, that are available in the community and in some pain clinics, may also be beneficial for patients with comorbid

hypertension and pain, including yoga, mindfulness, meditation, relaxation, Feldenkrais, or tai chi [60, 61]. While isolated interventions tend to only have a small impact on either blood pressure or pain, the greatest benefits have been found when implementing a multipronged approach comprising behavioural and pharmacological treatments together with bolstering social and clinical support [62]. Moreover, given the strong association between hypertension and other chronic conditions (i.e., diabetes and arthritis) providing a range of education and supportive care resources, action plans, psychological strategies and lifestyle advice may enhance treatment effectiveness by fostering greater self-management [62].

The present study provides some insight into the characteristics associated with hypertension in people living with persistent pain. However, further research is required to better understand the direction of the relationship between pain and hypertension using prospective study designs. In particular, we must examine whether particular pain-related illnesses or lifestyle behaviours play a causal or mediational role in the development and resolution of hypertension in the context of persistent pain. Given the association between pain severity and diseases like diabetes and hypertension, it may be that these conditions arise in part due to limited capacity of the heart to meet metabolic demands of everyday activities. This is particularly important given that cardiac reserve has important implications for the ability to respond to stress [63], mechanisms that are known to be impaired in several persistent pain conditions [13]. Moreover, cluster or network analyses may provide further insight into the patterns of demographic, lifestyle, illness and pain-related characteristics associated with hypertension. These analyses could be especially helpful in guiding the development of targeted interventions to better manage both persistent pain and hypertension.

## Limitations

Some limitations of this study should be considered. First, patients were only asked whether they had "high blood pressure", among a list of comorbid conditions, and clinical diagnoses or pharmacological management of those comorbidities were not recorded. While previous studies have shown that self-report of hypertension or high blood pressure is typically consistent with clinical diagnoses or treatment for hypertension based on electronic medical record review [64], some studies have found under-reporting of hypertension status in 8 to 10 percent of cases [65, 66]. Moreover, a recent meta-analysis showed that self-reported hypertension had only 42% sensitivity and 90% specificity for objectively assessed hypertension in epidemiological studies [67]. It is notable, however, that the data in that meta-analysis were heterogeneous. Given that the majority of studies under-estimated hypertension in self-reported data we speculate that the present study underestimates the prevalence of hypertension in pain clinic attendees compared with the general population, especially for patients who may not have been diagnosed with hypertension, forgot to report their diagnosis, were unwilling to disclose their diagnosis, or who had not had a recent blood pressure assessment [68]. To ensure better comparability with other epidemiological data sources such as the NHS it may be useful to modify the questions about comorbidities. Moreover, the associations between hypertension and persistent pain may be better understood through evaluation of blood pressure across the spectrum (i.e., as a continuous variable based on blood pressure recordings), rather than as a dichotomised characteristic of "normal" vs "high" blood pressure. This is particularly pertinent given that the curvilinear relationship between blood pressure and risk of major cardiovascular events [69]. The present findings should be replicated in studies that also objectively record blood pressure to confirm the presence of probable hypertension.

It is possible that hypertension prevalence between our sample and the comparison cohorts differed due to time-varying differences in population characteristics. Moreover, demographic

and clinical differences between the samples may partially explain these results, particularly given the increased risk ratios when accounting for age in the respective clinical populations. In ePPOC, several factors associated with hypertension are not recorded including smoking, alcohol consumption, physical activity levels, specific medications taken, or individual level socioeconomic measures such as education, occupation skill level, household income, or marital status. Nonetheless, neighbourhood level socioeconomic disadvantage was fairly consistently associated with higher odds of hypertension highlighting the potential role of social determinants in clinical characteristics. To enable analysis of health profiles and outcomes in people with persistent pain future studies could examine the role of smoking, physical activity or education.

We calculated BMI using previously recommended data cleaning criteria [39], and classified BMI according to the World Health Organization [20] cut-off ranges. While these are considered to be robust methods for cleaning large datasets and identifying people who were overweight or obese, we may have omitted some genuine BMI data for people in the extremely low or high weight and height ranges. Moreover, we acknowledge that BMI is not necessarily the best measure of overweight and obesity given that people with high lean mass could be classified in the same way as someone with high levels of body fat [70], and different cut-off points have been recommended for Asian populations [71]. Unfortunately implementation of alternate thresholds was not possible as our dataset only recorded country of birth and not ethnicity. Moreover, indices like percentage body fat, waist-to-hip ratio or waist circumference were not available. It should be noted that our sensitivity analyses revealed that the inclusion of all cases with imputation of missing data generated slightly more conservative odds ratios than if we had only included cases with complete data; however, these analyses did not have a substantial effect on which clinical or pain-related characteristics were associated with hypertension. Finally, the cross-sectional design of this study prevents us from making causal assumptions. The association between blood pressure and moderate to severe pain may be bidirectional [7] given that longitudinal studies show that cardiovascular disease is predictive of developing persistent pain [72, 73]. Prospective population cohort studies with objective assessment of blood pressure are required to gain insight into long-term causal mechanisms involved in comorbid pain and hypertension.

## Conclusions

For the first time we have documented the prevalence of hypertension in a large clinically representative cohort of patients attending tertiary pain management services in Australia. Several aspects of pain were associated with hypertension, particularly moderate to high pain severity. Most other characteristics associated with hypertension were consistent with life-stage and health-related mechanisms (i.e., advancing age, higher BMI and other comorbid conditions like arthritis and diabetes). Future studies should evaluate the effectiveness of treatments targeting factors associated with hypertension in patients with both pain and hypertension. Considering the population prevalence of persistent pain and cardiovascular diseases is expected to increase markedly over the next 30–40 years [74–76], the development of treatment protocols and guidelines for collaborative management of common comorbid chronic conditions will be essential to reducing the burden of those diseases.

## Supporting information

**S1 Table. Comparison of adjusted ORs for the imputed dataset (N = 43,789) and the cohort with complete data only (N = 21,653).**
(DOCX)

**S2 Table. Levels of pain interference in patients with and without hypertension.**
(DOCX)

# Acknowledgments

The electronic Persistent Pain Outcomes Collaboration (ePPOC) was initially established by the NSW Ministry of Health. We acknowledge all of the staff in the clinical services who collected and collated the data, and the feedback from the ePPOC Data Access Working Group and Scientific and Clinical Advisory Committee.

# Author Contributions

**Conceptualization:** Melita J. Giummarra, Andrew Tonkin, Carolyn A. Arnold.

**Data curation:** Melita J. Giummarra, Megan Blanchard.

**Formal analysis:** Melita J. Giummarra.

**Funding acquisition:** Melita J. Giummarra.

**Methodology:** Melita J. Giummarra, Hilarie Tardif, Megan Blanchard, Andrew Tonkin, Carolyn A. Arnold.

**Project administration:** Melita J. Giummarra.

**Resources:** Melita J. Giummarra, Hilarie Tardif, Megan Blanchard.

**Software:** Melita J. Giummarra.

**Writing – original draft:** Melita J. Giummarra.

**Writing – review & editing:** Melita J. Giummarra, Hilarie Tardif, Megan Blanchard, Andrew Tonkin, Carolyn A. Arnold.

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
