## [Decision Letter · Decision Letter 0]

19 Nov 2019

PONE-D-19-21769

Hypertension prevalence in patients attending tertiary pain management services, a registry-based Australian cohort study

PLOS ONE

Dear Dr. Giummarra,

Thank you for submitting your manuscript to PLOS ONE. After careful consideration, we feel that it has merit but does not fully meet PLOS ONE’s publication criteria as it currently stands. Therefore, we invite you to submit a revised version of the manuscript that addresses the points raised during the review process.

The reviewers address some points that should be considered. Please provide also a reply regarding availability of the data.

We would appreciate receiving your revised manuscript by Jan 03 2020 11:59PM. To enhance the reproducibility of your results, we recommend that if applicable you deposit your laboratory protocols in protocols.io, where a protocol can be assigned its own identifier (DOI) such that it can be cited independently in the future. For instructions see: http://journals.plos.org/plosone/s/submission-guidelines#loc-laboratory-protocols

We look forward to receiving your revised manuscript.

Kind regards,

Hans-Peter Brunner-La Rocca, M.D.

Academic Editor

PLOS ONE

Journal Requirements:

Reviewers' comments:

Reviewer's Responses to Questions

**Comments to the Author**

1. Is the manuscript technically sound, and do the data support the conclusions?

Reviewer #1: Yes

Reviewer #2: Yes

Reviewer #3: Yes

2. Has the statistical analysis been performed appropriately and rigorously? 

Reviewer #1: Yes

Reviewer #2: Yes

Reviewer #3: Yes

3. Have the authors made all data underlying the findings in their manuscript fully available?

Reviewer #1: No

Reviewer #2: Yes

Reviewer #3: Yes

4. Is the manuscript presented in an intelligible fashion and written in standard English?

Reviewer #1: Yes

Reviewer #2: Yes

Reviewer #3: Yes

5. Review Comments to the Author

Reviewer #1: Nice and clear paper, reads well and well explained. The finding of increased prevalence of hypertension in those visiting tertiary pain centers is interesting, but as this is the main finding, I feel it needs a bit more context where the comparison with the general population and GP patients is concerned;

In the methods section, the authors state that hypertension in the ePPOC was based on self-report. The NHS that was used for the general population was also based on self-report, but it remains unclear what the exact question was used for the NHS - some light is shed on it in the discussion, but that should be done much earlier. Similarly for the GP cohort, it is not clear from the methods on p7 how hypertension yes/no was defined here. Please specifiy this in the paper.

Furthermore, what is the explanation for the fact that hypertension in pain patients is much higher in the general population but not so much higher than in the GP group? Could this be just the fact that GP and pain clinic patients are generally less healthy, older etc? Nothing is said about this, it is only stated that there is another study with even higher prevalence - but we don't know whether this was self report as well, or not. The comparison with the other cohorts is relevant and has added value, but if it will be part of this paper it needs to be properly discussed.

- page 7 statistical analysis - the weight criteria (and height, to lesser extent) used to define whether the data entered was valid or not seem rather extreme, would a man said to weigh 35 kg really be included as such in the BMI calculations? Probably there were no such cases in the dataset but this sentence makes one wonder what could have gone wrong in the analyses.

- the protective effect of depressive symptoms is rather surprising. Has this ever been described before? Would there be an explanation?

Minor textual revisions:

- Abstract second last sentence: 'Hypertension was more prevalent in people with persistant pain, is associated severe pain' - this should probably be 'associated WITH severe pain'

- p13 - below middle - 'a recent study recent found' - the last recent should be deleted

- p16 'this particularly pertinent' - this IS particularly pertinent?

- p17 conlusion last sentence 'will be integral' - I would expect something like vital or essential instead of integral but that may not be what was intended.

Reviewer #2: Giummarra and colleagues aimed to evaluate the prevalence of hypertension in a population suffering from chronic pain. For this purpose they evaluated the electronic Persistent Pain Outcomes Collaboration Registry between 2013 and 2018. They show that patients with chronic pain suffer more frequently from hypertension. Moreover, pain severity was independently associated with hypertension after adjustment for covariates.

The main limitation of the study relies on the self-reporting methods used to evaluate medical conditions and comorbidities. The fact that patients are not always aware of the medical conditions they suffer from, may introduce an important bias in the analysis.

Furthermore, the covariates that has been used to calculate the adjusted model should be mentioned in the statistical analysis section.

The authors have chosen to impute covariates. Why perform imputation? Is there any baseline difference between the imputed and non-imputed data?

Baseline characteristics show total and hypertensive population, why is there the non-hypertensive population not included? Supplementary table 1 shows this difference for a very limited amount of variables, but would be interesting to evaluate the differences in all studies variables and to add it in an additional supplementary table.

A visual evaluation of the main variables (i.e. pain severity and other comorbidities like Diabetes Mellitus and arthritis) and hypertension would be very illustrative and would be helpful to understand a quite complex statistical analysis in a single shoot. For this purpose, I would suggest to include a Network analysis figure.

It is remarkable that specially illness related pain and pain severity are related with hypertension. Could be related with underlying diseases and therefor poorer cardiovascular reserve? In this that this issue should be broader discussed in the discussion. Moreover, a mediation analysis could be very helpful to elucidate the direction of this relationships.

Reviewer #3: The paper of Guimmarra et al investigates 1) the prevalence of hypertension and 2) the factors associated with hypertension in patients in tertiary pain clinics. To this end, the Persistent Pain Outcomes Collaboration registry was used (n=43,789). 23.9% had self-reported hypertension. Factors associated with self-reported hypertension were higher age, lower socioeconomic status, higher BMI, born outside Australasia, comorbid arthritis, diabetes or severe anxiety symptoms. Protective factors were female sex and depressive symptoms. The authors suggest that screening for hypertension in pain clinics may improve treatment outcomes.

Several points may be addressed:

1. What was the rationale to categorize variables such as age, BMI and other scores?

In addition, for BMI, there may be some misclassification, as the cutoff point for overweight is different for Asians.

2. Have the authors considered multiple testing and ways to correct for this (e.g. false discovery rate?)

3. Can authors reflect on what the potential consequence may be of using only self-reported hypertension (e.g. underestimation or overestimation of the effect sizes?)

4. Authors correctly state in the limitations that the higher hypertension prevalence in the ePOCC sample compared with the general population/primary care samples may be due to demographic and clinical differences. Do the authors have information about covariates in the general population sample and primary care sample? Otherwise, authors adjust for such differences and present adjusted RRs, which is more informative.

5. Many previous studies suggest that higher blood pressure is associated with depression, as opposed to ‘no hypertension’. Was this finding a false positive finding? Or is there a biological mechanism to this phenomenon?

6. Table 1 may be more informative if the ‘no hypertension’ group was described as well (authors may consider to omit the ‘total’ group). I understand that it can be calculated by the reader, but it is a hassle.

7. On page 15, first line, BP was used as an abbreviation, but was not defined earlier. Please check all abbreviations if they were defined.

6. PLOS authors have the option to publish the peer review history of their article (what does this mean?). If published, this will include your full peer review and any attached files.

Reviewer #1: No

Reviewer #2: No

Reviewer #3: Yes: Tan Lai Zhou

---

## [Author Response · Author response to Decision Letter 0]

19 Dec 2019

Dear Professor Hans-Peter Brunner-La Rocca,

We thank you for the opportunity to revise and resubmit our manuscript. The reviewers have made some excellent suggestions, which we have taken on board in our revision of the manuscript. We respond below, point by point, to each comment and indicate where the respective changes have been made to the track change version of the manuscript.

On behalf of my co-authors,

Dr Melita Giummarra

Reviewer #1: Nice and clear paper, reads well and well explained. 

The finding of increased prevalence of hypertension in those visiting tertiary pain centers is interesting, but as this is the main finding, I feel it needs a bit more context where the comparison with the general population and GP patients is concerned. In the methods section, the authors state that hypertension in the ePPOC was based on self-report. The NHS that was used for the general population was also based on self-report, but it remains unclear what the exact question was used for the NHS - some light is shed on it in the discussion, but that should be done much earlier. Similarly for the GP cohort, it is not clear from the methods on p7 how hypertension yes/no was defined here. Please specify this in the paper.

RESPONSE: Thank you for this suggestion. We have described the data collection procedures for the NHS and primary care study in the Methods (Page 6-7). To reduce duplication we have moved the NHS interview questions from the Discussion to the Methods section.

Furthermore, what is the explanation for the fact that hypertension in pain patients is much higher in the general population but not so much higher than in the GP group? Could this be just the fact that GP and pain clinic patients are generally less healthy, older etc? Nothing is said about this, it is only stated that there is another study with even higher prevalence - but we don't know whether this was self report as well, or not. The comparison with the other cohorts is relevant and has added value, but if it will be part of this paper it needs to be properly discussed.

RESPONSE: This is an excellent point, which we now discuss further in the Discussion on page 14. 

Page 7 statistical analysis - the weight criteria (and height, to lesser extent) used to define whether the data entered was valid or not seem rather extreme, would a man said to weigh 35 kg really be included as such in the BMI calculations? Probably there were no such cases in the dataset but this sentence makes one wonder what could have gone wrong in the analyses.

RESPONSE: We understand the reviewers concern; however, we were not aware of any other criteria to apply during data cleaning to ensure that our BMI data were valid. As ePPOC relies on patients and clinicians to enter data, it appears that in some cases height and weight data have been entered erroneously. Upon screening the data we identified that the most common errors were probably that these data were entered into the incorrect field (i.e., weight in the height field, and vice versa), or that height and weight were entered in a metric other than kilograms or metres. 

Given that this was such a large dataset we needed to apply a systematic rule to omit these data entry errors and probable invalid height and weight combinations. We therefore elected to use criteria that had previously been used in the Australian epidemiological study. We acknowledge this as a limitation on Page 18.

The protective effect of depressive symptoms is rather surprising. Has this ever been described before? Would there be an explanation?

RESPONSE: We have further discussed this finding in the context of the literature on the association between depression and hypertension on Page 15.

Minor textual revisions:

Abstract second last sentence: 'Hypertension was more prevalent in people with persistent pain, is associated severe pain' - this should probably be 'associated WITH severe pain'.

RESPONSE: This part of the abstract was highlighting that the prevalence was higher in our cohort with persistent pain than in the general community. To make this clearer we’ve revised the wording as follows:

“Hypertension was more prevalent in people with persistent pain than in the general community, was associated with more severe pain, and commonly co-occurred with pain-related impairments”

p13 - below middle - 'a recent study recent found' - the last recent should be deleted

RESPONSE: Thank you, we have deleted the second “recent”.

p16 'this particularly pertinent' - this IS particularly pertinent?

RESPONSE: Thank you, we have made this correction.

p17 conclusion last sentence 'will be integral' - I would expect something like vital or essential instead of integral but that may not be what was intended.

RESPONSE: This is a great suggestion. We have rephrased this expression to “will be essential to reducing...” as suggested.

Reviewer #2

Giummarra and colleagues aimed to evaluate the prevalence of hypertension in a population suffering from chronic pain. For this purpose they evaluated the electronic Persistent Pain Outcomes Collaboration Registry between 2013 and 2018. They show that patients with chronic pain suffer more frequently from hypertension. Moreover, pain severity was independently associated with hypertension after adjustment for covariates.

The main limitation of the study relies on the self-reporting methods used to evaluate medical conditions and comorbidities. The fact that patients are not always aware of the medical conditions they suffer from, may introduce an important bias in the analysis.

RESPONSE: We agree that this is the key limitation of the study, which is why we discuss it extensively in the discussion on Page 17-18. We are not sure what else we can add to further highlight this cautionary point.

Furthermore, the covariates that has been used to calculate the adjusted model should be mentioned in the statistical analysis section.

RESPONSE: The covariates are now listed in the statistical analysis section on Page 8.

The authors have chosen to impute covariates. Why perform imputation? Is there any baseline difference between the imputed and non-imputed data?

RESPONSE: We chose to impute missing data for the multivariable analysis so that the analyses included the total cohort, and not the subset of patients with 100% complete data. 

For most variables missing data was present for approximately 4-5% of the cohort. The exceptions were that 12% did not indicate whether they had diabetes, and 26% did not have a BMI due to failure to report height or weight, or due to data entry errors resulting in omission of the BMI from analyses. The levels of missing data are reported in the footnote of Table 1.

To examine whether the effects were different in the whole cohort versus those with complete data we now include a “complete cases” analysis in supplementary materials. The imputed dataset did not yield substantially different effects to the complete cases cohort, but we do note in the results that the imputed dataset yielded smaller AORs than the complete dataset (Page 12-13). We feel that it is more appropriate to err on the side of caution and to not overstate the potential relationship between pain and hypertension, and therefore retain the imputed dataset for our primary analyses. We acknowledge these sensitivity analyses in the discussion on Page 19 with the following sentence:

“It should be noted that our sensitivity analyses revealed that the inclusion of all cases with imputation of missing data generated slightly more conservative odds ratios than if we had only included cases with complete data; however, these analyses did not have a substantial effect on which clinical or pain-related characteristics were associated with hypertension.”

Baseline characteristics show total and hypertensive population, why is there the non-hypertensive population not included? Supplementary table 1 shows this difference for a very limited amount of variables, but would be interesting to evaluate the differences in all studies variables and to add it in an additional supplementary table.

RESPONSE: We have added the number (%) of people in the non-hypertensive group with each of the “predictor” characteristics, see Table 1 column 3.

A visual evaluation of the main variables (i.e. pain severity and other comorbidities like Diabetes Mellitus and arthritis) and hypertension would be very illustrative and would be helpful to understand a quite complex statistical analysis in a single shoot. For this purpose, I would suggest to include a Network analysis figure.

RESPONSE: Conducting a network or cluster analysis on these data does sound like a very interesting idea. However, we do not believe that this is necessary to address the aims of the present study: to identify which demographic, clinical and pain-related characteristics are associated with having hypertension in people living with persistent and disabling pain. Moreover, as multicollinearity was not problematic in our data, and we are not concerned about the relationship between the main “predictor” variables. 

That said, we do wish to highlight the potential utility of these approaches in future research to allow us to better understand factors associated with having hypertension in the context of persistent pain. Therefore, we have added further discussion about the potential use of cluster and network analyses in future research in the Discussion (see the bottom of the new paragraph on Page 17).

It is remarkable that specially illness related pain and pain severity are related with hypertension. Could be related with underlying diseases and therefor poorer cardiovascular reserve? In this that this issue should be broader discussed in the discussion. Moreover, a mediation analysis could be very helpful to elucidate the direction of this relationships.

RESPONSE: We really like the suggestion that cardiac reserve may be impaired in these patients, which we now briefly discuss on Page 17. As the study data are cross-sectional we are cautious about the fact that a mediation analysis might provide misleading insight into potential directionality of the relationship between hypertension and pain severity. However, future prospective research could certainly address this potential question, which we have now suggested in the discussion on Page 17.

Reviewer #3

The paper of Guimmarra et al investigates 1) the prevalence of hypertension and 2) the factors associated with hypertension in patients in tertiary pain clinics. To this end, the Persistent Pain Outcomes Collaboration registry was used (n=43,789). 23.9% had self-reported hypertension. Factors associated with self-reported hypertension were higher age, lower socioeconomic status, higher BMI, born outside Australasia, comorbid arthritis, diabetes or severe anxiety symptoms. Protective factors were female sex and depressive symptoms. The authors suggest that screening for hypertension in pain clinics may improve treatment outcomes.

Several points may be addressed:

1. What was the rationale to categorize variables such as age, BMI and other scores? In addition, for BMI, there may be some misclassification, as the cutoff point for overweight is different for Asians.

RESPONSE: Thank you for these comments. We chose to use categorical predictors to allow us to identify the differences in odds of having hypertension according to the level of the predictor variable, which does not always vary linearly across the range of possible values. We used recommended or validated cut-points for BMI (i.e., the WHO recommended cu-points) and all questionnaires, and commonly used categories for age, to allow comparison with other studies. We now acknowledge the potential misclassification of some people on the BMI weight ranges based on lean mass or ethnicity in the Discussion (see Page 18-19).

2. Have the authors considered multiple testing and ways to correct for this (e.g. false discovery rate?)

RESPONSE: Thank you for this comment. For the analyses examining the relative risk of having hypertension there were only three analyses. While we report the 95% CIs in the paper, we can report that the p-values were <0.0001 for each of these contrasts, and so they would hold up if adjusting for multiple comparisons (e.g., using the Benjamini-Hochberg procedure) for false discovery rates. Likewise, the results from the t-tests examining differences in pain interference also hold up even with correction for multiple comparisons when applying corrections for false-discovery rate as the p-values are already very low, but we do note in the paper that these are probably not clinically meaningful differences given the small effect sizes.

The primary analyses are the multivariable logistic regression it is not necessary to make corrections for multiple testing when examining factors associated with hypertension. It is routine to report the unadjusted odds when conducting multivariable analyses, but these are not the focus of the analyses. 

As a result of these considerations we have not made any changes to the manuscript regarding corrections for false discovery rate.

3. Can authors reflect on what the potential consequence may be of using only self-reported hypertension (e.g. underestimation or overestimation of the effect sizes?)

RESPONSE: We have suggested that our study probably indicates an underestimation of the prevalence of hypertension (page 17, “therefore we speculate that the present study underestimates the prevalence of hypertension in pain clinic attendees compared with the general population”). Our data probably underestimate the relative risk of pain clinic attendees having hypertension compared with the general population, and includes some measurement error in the magnitude of the relationship between demographic, health and pain-related characteristics and hypertension. However, as we cannot be sure which patients underreported their hypertension status to err on the side of caution we do not speculate on the effect of under-reporting on these latter relationships.

4. Authors correctly state in the limitations that the higher hypertension prevalence in the ePOCC sample compared with the general population/primary care samples may be due to demographic and clinical differences. Do the authors have information about covariates in the general population sample and primary care sample? Otherwise, authors adjust for such differences and present adjusted RRs, which is more informative.

RESPONSE: Unfortunately, the NHS reports did not provide age- or covariate-adjusted prevalence rates; however, the sentinel study did provided crude and age-adjusted prevalence of hypertension. We have compared these rates with the age-adjusted prevalence of hypertension in the ePPOC cohort and found an even greater relative risk of hypertension in the ePPOC cohort when accounting for age. These new analyses are reported on Page 11-12, and noted in the first paragraph of the discussion on Page 14.

5. Many previous studies suggest that higher blood pressure is associated with depression, as opposed to ‘no hypertension’. Was this finding a false positive finding? Or is there a biological mechanism to this phenomenon?

RESPONSE: We have further discussed this finding in the context of the literature on the association between depression and hypertension on Page 15.

6. Table 1 may be more informative if the ‘no hypertension’ group was described as well (authors may consider to omit the ‘total’ group). I understand that it can be calculated by the reader, but it is a hassle.

RESPONSE: We have added this column to the Table.

7. On page 15, first line, BP was used as an abbreviation, but was not defined earlier. Please check all abbreviations if they were defined.

RESPONSE: We have changed this to the full expression given that we do not use the same abbreviation again. We have also checked that all other abbreviations are defined appropriately.

---

## [Decision Letter · Decision Letter 1]

9 Jan 2020

Hypertension prevalence in patients attending tertiary pain management services, a registry-based Australian cohort study

PONE-D-19-21769R1

Dear Dr. Giummarra,

We are pleased to inform you that your manuscript has been judged scientifically suitable for publication and will be formally accepted for publication once it complies with all outstanding technical requirements.

With kind regards,

Hans-Peter Brunner-La Rocca, M.D.

Academic Editor

PLOS ONE

Additional Editor Comments (optional):

Reviewers' comments:

Reviewer's Responses to Questions

**Comments to the Author**

1. If the authors have adequately addressed your comments raised in a previous round of review and you feel that this manuscript is now acceptable for publication, you may indicate that here to bypass the “Comments to the Author” section, enter your conflict of interest statement in the “Confidential to Editor” section, and submit your "Accept" recommendation.

Reviewer #2: All comments have been addressed

Reviewer #3: All comments have been addressed

2. Is the manuscript technically sound, and do the data support the conclusions?

Reviewer #2: Yes

Reviewer #3: Yes

3. Has the statistical analysis been performed appropriately and rigorously? 

Reviewer #2: Yes

Reviewer #3: Yes

4. Have the authors made all data underlying the findings in their manuscript fully available?

Reviewer #2: Yes

Reviewer #3: Yes

5. Is the manuscript presented in an intelligible fashion and written in standard English?

Reviewer #2: Yes

Reviewer #3: Yes

6. Review Comments to the Author

Reviewer #2: The authors addressed correctly all comments and improved the manuscript considerably. Supplemental material has been added and the main statistical and methodological issues have been tackled.

Reviewer #3: The authors have sufficiently addressed the comments and the manuscript has ameliorated, I have no further suggestions.

7. PLOS authors have the option to publish the peer review history of their article (what does this mean?). If published, this will include your full peer review and any attached files.

Reviewer #2: Yes: Arantxa Barandiaran Aizpurua

Reviewer #3: Yes: Tan Lai Zhou

---

## [Editor Report · Acceptance letter]

15 Jan 2020

PONE-D-19-21769R1 

Hypertension prevalence in patients attending tertiary pain management services, a registry-based Australian cohort study 

Dear Dr. Giummarra:

I am pleased to inform you that your manuscript has been deemed suitable for publication in PLOS ONE. Congratulations! Your manuscript is now with our production department. 

With kind regards,

on behalf of

Dr. Hans-Peter Brunner-La Rocca 

Academic Editor

PLOS ONE